# The Ca$^{2+}$-activated cation channel TRPM4 is a positive regulator of pressure overload-induced cardiac hypertrophy

Yang Guo[1,2,3†], Ze-Yan Yu[1,2,3†], Jianxin Wu[1], Hutao Gong[2], Scott Kesteven[2,3], Siiri E Iismaa[1,3], Andrea Y Chan[1], Sara Holman[1], Silvia Pinto[4,5], Andy Pironet[4,5], Charles D Cox[1,3], Robert M Graham[1,3], Rudi Vennekens[4,5], Michael P Feneley[2,3,6*], Boris Martinac[1,3*]

[1]Molecular Cardiology and Biophysics Division, Victor Chang Cardiac Research Institute, Sydney, Australia; [2]Cardiac Physiology and Transplantation Division, Victor Chang Cardiac Research Institute, Sydney, Australia; [3]St Vincent's Clinical School, Faculty of Medicine, University of New South Wales, Sydney, Australia; [4]Laboratory of Ion Channel Research, Department of Molecular and Cellular Medicine, Katholieke Universiteit Leuven, Leuven, Belgium; [5]TRP Research Platform Leuven (TRPLe), Katholieke Universiteit Leuven, Leuven, Belgium; [6]Department of Cardiology, St Vincent's Hospital, Sydney, Australia

*For correspondence:
michael.feneley@svha.org.au (MPF);
b.martinac@victorchang.edu.au (BM)

†These authors contributed equally to this work

Competing interests: The authors declare that no competing interests exist.

**Abstract** Pathological left ventricular hypertrophy (LVH) occurs in response to pressure overload and remains the single most important clinical predictor of cardiac mortality. The molecular pathways in the induction of pressure overload LVH are potential targets for therapeutic intervention. Current treatments aim to remove the pressure overload stimulus for LVH, but do not completely reverse adverse cardiac remodelling. Although numerous molecular signalling steps in the induction of LVH have been identified, the initial step by which mechanical stretch associated with cardiac pressure overload is converted into a chemical signal that initiates hypertrophic signalling remains unresolved. In this study, we show that selective deletion of transient receptor potential melastatin 4 (TRPM4) channels in mouse cardiomyocytes results in an approximately 50% reduction in the LVH induced by transverse aortic constriction. Our results suggest that TRPM4 channel is an important component of the mechanosensory signalling pathway that induces LVH in response to pressure overload and represents a potential novel therapeutic target for the prevention of pathological LVH.

## Introduction

Pathological left ventricular hypertrophy (LVH) is the most powerful independent predictor for cardiovascular mortality (*Levy et al., 1990*; *Mudd and Kass, 2008*). It occurs in response to two very common clinical conditions: systemic hypertension and aortic valve stenosis. It manifests as increased cardiomyocyte volume and weight (*Frey et al., 2004*; *Nicks et al., 2020*; *Swynghedauw, 1999*; *Wilde and Brugada, 2011*), which results in increased heart mass, particularly left ventricular (LV) mass. Although pathological LVH commonly occurs as a response to increased cardiac wall stress, sometimes termed 'compensatory hypertrophy', it is now well established that the effects of pathological LVH are deleterious for heart function, leading to increased cardiac failure and death (*Levy et al., 1990*; *Mudd and Kass, 2008*). So far, the only treatment for this condition is lowering elevated blood pressure or replacing a stenotic aortic valve. However, these treatments cannot completely reverse the pathological effects on the myocardium once LVH is established. Consequently, understanding the molecular mechanisms underlying pathological LVH may lead to

therapies directed at preventing, inhibiting, or reversing pathological LVH and reducing its associated morbidity and mortality.

The development of pathological LVH depends on upstream stimuli, such as mechanical forces (e.g. pressure overload) or neuroendocrine hormones (e.g. angiotensin II), and distinct downstream signalling mechanisms (*Maillet et al., 2013*; *Nakamura and Sadoshima, 2018*; *Saucerman et al., 2019*; *Tham et al., 2015*; *van Berlo et al., 2013*). Importantly, a large body of work implicates intracellular $Ca^{2+}$ levels and subsequent activation of $Ca^{2+}$/calmodulin (CaM)-dependent signalling pathways, such as the calcineurin-nuclear factor of activated T-cells (NFAT)-GATA4 axis, in the induction of pathological LVH (*Bers, 2008*; *Kehat and Molkentin, 2010*; *Molkentin, 2000*; *Molkentin, 2013*; *Molkentin et al., 1998*; *Wilkins et al., 2004*; *Wilkins and Molkentin, 2004*; *Zarain-Herzberg et al., 2011*). Gq-coupled receptors are thought to play an important role in the induction of pathological LVH in response to both neurohumoral stimulation (*Adams et al., 1998*; *Nakamura and Sadoshima, 2018*; *Paradis et al., 2000*) and mechanical forces, such as the increase in LV afterload induced by experimental aortic constriction (*Keys et al., 2002*). Once activated, Gq-coupled receptors are thought to then activate the calcineurin-NFAT pathway (*Kehat and Molkentin, 2010*; *Molkentin, 2000*; *Molkentin, 2013*; *Molkentin et al., 1998*; *Wilkins et al., 2004*; *Wilkins and Molkentin, 2004*).

Our previous experimental work, however, has demonstrated that although Gq-coupled receptors and the calcineurin-NFAT pathway are essential for the induction of LVH in response to angiotensin II, neither are required for the induction of LVH in response to transverse aortic constriction (TAC), the most common experimental model of LV pressure overload (*Yu et al., 2021*), and one not associated with activation of the renin–angiotensin system (*Zou et al., 2004*). In contrast to the lack of activation of the calcineurin-NFAT pathway with TAC, an alternative $Ca^{2+}$/CaM-dependent signalling pathway, the $Ca^{2+}$/CaM-dependent protein kinase II (CaMKII)-histone deacetylase (HDAC)-myocyte enhancer factor 2 (MEF2) pathway (*Backs et al., 2009*; *Backs et al., 2006*; *Passier et al., 2000*), is activated in response to TAC (*Yu et al., 2021*).

Left unexplained by our previous work, however, is the mechanism by which the CaMKII-HDAC-MEF2 pathway is activated by TAC, given that this activation is not dependent on Gq-coupled receptors. Prime candidates for mediating this mechanism are mechanosensitive ion channels. In cardiac mechanotransduction, where mechanical stimuli are converted into electrical or chemical signals (*Martinac and Cox, 2017*; *Peyronnet et al., 2016*), $Ca^{2+}$-dependent ion channels, such as transient receptor potential (TRP) channels, act as important modulators of intracellular $Ca^{2+}$ homeostasis (*Wang et al., 2018*) and are thought to be unique biosensors that activate specific pathological LVH signalling pathways (*Eder and Molkentin, 2011*; *Wu et al., 2010*). As a $Ca^{2+}$- and voltage-activated non-selective monovalent cation channel, TRP cation channel subfamily melastatin 4 (TRPM4) may contribute to an increase in intracellular $Ca^{2+}$ concentration by causing membrane depolarisation (*Launay et al., 2002*), although we and others (*Constantine et al., 2016*; *Gottlieb et al., 2008*; *Nikolaev et al., 2019*) have demonstrated that mammalian TRP channels, including TRPM4, are not directly activated by membrane stretch. Consequently, if TRPM4 plays a role in TAC-induced LVH, it acts as an amplifier of the primary $Ca^{2+}$ or voltage signal from a yet to be determined mechanosensitive ion channel or channels. TRPM4 has been functionally characterised in atrial and ventricular cardiomyocytes, both human and rodent (*Guinamard et al., 2004*; *Guinamard et al., 2006*; *Watanabe et al., 2008*). Other studies indicate that TRPM4 contributes to both cardiac function and disease development, including cardiac hypertrophy and heart failure (*Frede et al., 2020*; *Gueffier et al., 2017*; *Hedon et al., 2021*; *Jacobs et al., 2015*; *Mathar et al., 2014*). Previous studies using *Trpm4* cardiomyocyte-specific knock-out (*Trpm4* cKO) mice have shown that TRPM4 is a negative regulator of angiotensin II-induced cardiac hypertrophy in mice, which involves the calcineurin-NFAT pathway (*Kecskés et al., 2015*), and that TRPM4 is essential for survival after myocardial infarction (*Hedon et al., 2021*; *Jacobs et al., 2015*). However, whether TRPM4 plays a role in mechanical pressure overload-induced LVH has yet to be determined.

Here, we investigate the role of TRPM4 in pressure overload LVH induced by TAC in homozygous *Trpm4* cKO mice (*Kecskés et al., 2015*) as compared to wild-type (WT) control mice. We demonstrate that loss of cardiomyocyte TRPM4 significantly attenuates the development of LVH observed in response to TAC in WT mice. Moreover, this effect is associated with reduced activation of the CaMKII-HDAC4-MEF2 pathway.

## Results

### Development of LV hypertrophy in response to pressure overload at 14 days after TAC in WT mice

As documented in our previous study (*Yu et al., 2021*), TAC-induced cardiac hypertrophy is a response to LV pressure overload. As expected, LV systolic pressure increased by ~65 mmHg (p<0.001) 14 days after TAC (*Table 1*), whereas heart rate (*Table 1*), $dP/dt_{max}$, and $dP/dt_{min}$ (*Table 1*) remained unaltered. Consistent with 14 days of TAC resulting in a compensated LVH model, body weight (BW) and lung weight (LW) (*Table 1*) remained unchanged in TAC mice compared to sham-operated mice. These results indicate that our TAC model at 14 days remains an excellent model of compensated LVH rather than heart failure.

Representative photos illustrate the size differences of WT mouse hearts after 2 days and 14 days of sham or TAC (*Figure 1—figure supplement 1A*). LVH was not detected 2 days after TAC (*Table 1*, *Figure 1—figure supplement 1A*), whereas we observed significant LVH in WT mice 14 days after TAC, as evidenced by increases in heart weight (HW, p<0.001), left ventricular weight (LVW, p<0.001), and the ratios of heart weight/body weight (HW/BW, p<0.001), left ventricle weight/body weight (LVW/BW, p<0.001), and left ventricle weight/tibial length (LVW/TL, p<0.001), without changes in BW or TL, when compared with the sham-operated animals (*Table 1*). Consistent with the development of pathological hypertrophy, TAC was associated with an increase in cardiac fibrosis (p<0.001, *Table 1*, *Figure 1—figure supplement 1B*) and enhanced collagen III (*Col3a1*) expression (p<0.001, *Table 1*).

Haemodynamic parameters were measured in wild-type (WT) mice 14 days after being subjected to TAC versus sham-operated controls (n = 7–11/group). Post-mortem analysis of mice 2 days or 14 days after sham or TAC; LVH developed 14 days after TAC, indicated by the ratios of HW/BW, LVW/BW, and LVW/TL in WT mice subjected to TAC versus sham-operated controls. Cardiac fibrosis was evaluated by Masson's trichrome staining of LV tissue from WT mice subjected to 2 days or 14 days of TAC versus sham-operated controls; cardiac fibrosis areas were graded (n = 5–6/group). Relative Collagen III (*Col3a1*) mRNA expression was normalised by GAPDH and calculated as fold change relative to sham in 2 days and 14 days groups, respectively (n = 4/group). LVSP: left ventricular systolic pressure; HR: heart rate; dP/dt: first derivative of pressure with respect to time; BW: body weight; HW: heart weight; LVW: left ventricular weight; LW: lung weight; TL: tibia length; HW/BW: heart weight to body weight ratio; LVW/BW: LV weight to body weight ratio; LVW/TL: LV weight to tibia length ratio; LW/BW: lung weight to body weight ratio. Results are presented as means ± SEM. **p<0.01, ***p<0.001, compared between sham- and TAC-operated groups.

### Early gene markers of induction of pathological hypertrophy in WT mice

Although there was no significant LVH 2 days after TAC (*Table 1*, *Figure 1—figure supplement 1A*), induction of hypertrophy-associated genes (atrial natriuretic peptide [ANP, *Nppa*; 9.9-fold, p<0.01], brain natriuretic peptide [BNP, *Nppb*; 8.1-fold, p<0.01], and α-skeletal actin [α-SA, *Acta1*; 4.5-fold, p<0.01]) was already evident at this time (*Table 2*), and expression of these genes remained high at 14 days (ANP, *Nppa*; p<0.001, BNP, *Nppb*; p<0.001, and α-SA, *Acta1*; p<0.001,*Table 2*).

### TRPM4 expression was downregulated in response to LV pressure overload in WT mice

To examine whether the TRPM4 ion channel is involved in TAC-induced LVH, we conducted real-time quantitative PCR (RT-PCR) on LV tissues or isolated LV cardiomyocytes from TAC- or sham-operated hearts. *Trpm4* mRNA expression in LV tissue (*Figure 1A*) and isolated cardiomyocytes (*Figure 1B*) fell by 50% (p<0.001) and 57% (p<0.001), respectively, in response to 2 days of TAC, and expression continued to be reduced by 30% (p<0.05, *Figure 1A*) and 40% (p<0.001, *Figure 1B*), respectively, at 14 days. Consistent with the mRNA changes, TRPM4 protein expression in LV tissue and isolated cardiomyocytes also fell significantly, particularly in cardiomyocytes, after 14 days of TAC (p<0.001) (*Figure 1C–E*).

**Table 1.** Haemodynamic parameters were measured in wild-type (WT) mice 14 days after subjected to TAC versus sham-operated controls (n = 7–11/group).

Post-mortem analysis of mice 2 days or 14 days after sham or TAC; LVH developed 14 days after TAC, indicated by the ratios of HW/BW, LVW/BW, and LVW/TL in WT mice subjected to TAC versus sham-operated controls. Cardiac fibrosis was evaluated by Masson's trichrome staining of LV tissue from WT mice subjected to 2 days or 14 days of TAC versus sham-operated controls; cardiac fibrosis areas were graded (n = 5–6/group). Relative Collagen III (Col3a1) mRNA expression was normalised by GAPDH and calculated as fold change relative to sham in 2 days and 14 days groups, respectively (n = 4/group). LVSP: left ventricular systolic pressure; HR: heart rate; dP/dt: first derivative of pressure with respect to time. BW: body weight; HW: heart weight; LVW: left ventricular weight; LW: lung weight; TL: tibia length; HW/BW: heart weight to body weight ratio; LVW/BW: LV weight to body weight ratio; LVW/TL: LV weight to tibia length ratio; LW/BW: lung weight to body weight ratio. Results are presented as means ± SEM. **p<0.01, ***p<0.001, compared between sham- and TAC-operated groups.

| | 2 days | | 14 days | |
| --- | --- | --- | --- | --- |
| | Sham | TAC | Sham | TAC |
| Haemodynamic parameter | | | | |
| n | | | 7 | 7 |
| HR (bpm) | | | 506 ± 4 | 506 ± 3 |
| Aortic systolic pressure (mmHg) | | | 103 ± 1 | 164 ± 2*** |
| Aortic diastolic pressure (mmHg) | | | 76 ± 1 | 74 ± 1 |
| LV systolic Pressure (mmHg) | | | 105 ± 3 | 164 ± 8*** |
| dP/dt$_{max}$ (mmHg/s) | | | 9438 ± 367 | 9838 ± 259 |
| dP/dt$_{min}$ (mmHg/s) | | | −9666 ± 377 | −10108 ± 364 |
| Anatomical parameter | | | | |
| n | 8 | 8 | 11 | 11 |
| BW (g) | 28.5 ± 0.3 | 27.7 ± 0.5 | 28.6 ± 0.3 | 27.2 ± 0.5 |
| HW (mg) | 136.7 ± 2.2 | 132.8 ± 1.3 | 133.1 ± 1.9 | 176.1 ± 3.6 *** |
| LVW (mg) | 98.0 ± 2.0 | 97.7 ± 1.4 | 96.4 ± 1.8 | 136.1 ± 1.4 *** |
| LW (mg) | 141.9 ± 0.9 | 143.6 ± 1.5 | 146.9 ± 1.8 | 147.0 ± 1.9 |
| TL (mm) | 17.4 ± 0.1 | 17.5 ± 0.2 | 17.5 ± 0.2 | 17.2 ± 0.1 |
| HW/BW (mg/g) | 4.8 ± 0.1 | 4.8 ± 0.1 | 4.6 ± 0.1 | 6.6 ± 0.1 *** |
| LVW/BW (mg/g) | 3.4 ± 0.1 | 3.5 ± 0.1 | 3.4 ± 0.1 | 5.1 ± 0.1 *** |
| LVW/TL (mg/mm) | 5.6 ± 0.1 | 5.6 ± 0.1 | 5.3 ± 0.1 | 7.9 ± 0.1 *** |
| LW/BW (mg/g) | 5.0 ± 0.1 | 5.2 ± 0.1 | 5.2 ± 0.1 | 5.4 ± 0.1 |
| **Assessment of cardiac fibrosis** | | | | |
| n | 5 | 5 | 6 | 6 |
| Fibrosis areas (%) | 4.0 ± 0.2 | 3.6 ± 0.2 | 4.4 ± 0.1 | 12.4 ± 0.5*** |
| n | 4 | 4 | 4 | 4 |
| Collagen III mRNA expression (fold change) | 1.0 ± 0.1 | 5.7 ± 0.8** | 1.0 ± 0.1 | 5.1 ± 0.7** |

The online version of this article includes the following source data for Table 1:

Source data 1. Haemodynamic and anatomical parameters.

## TRPM4 deficiency decreases the hypertrophic response to TAC-induced pressure overload

To further investigate the role of TRPM4 channels in pressure overload-induced LVH, we performed TAC or sham surgery in mice with cardiomyocyte-specific, conditional deletion of *Trpm4* (*Trpm4* cKO) using Cre expression driven by the *Myl7* promoter (*Kecskés et al., 2015*). Results obtained in

**Table 2.** Early markers of LVH induction in response to left ventricular pressure overload in WT mice. Relative mRNA expression of ANP (Nppa), BNP (Nppb), and a-SA (Acta1) after 2 days or 14 days of TAC compared to sham (n = 4–5/group). The relative mRNA expression was normalised by GAPDH and calculated as fold change relative to sham in 2 days and 14 days groups, respectively.Results are presented as means ± SEM. **p<0.01, ***p<0.001, compared between sham- and TAC-operated groups.

| | 2 days | | 14 days | |
|---|---|---|---|---|
| | Sham | TAC | Sham | TAC |
| LVH markers (fold change) | | | | |
| n | 4 | 4 | 5 | 5 |
| ANP | 1.0 ± 0.1 | 9.9 ± 1.1** | 1.0 ± 0.1 | 9.6 ± 0.7*** |
| BNP | 1.0 ± 0.1 | 8.1 ± 0.8** | 1.0 ± 0.2 | 7.5 ± 0.4*** |
| α-SA | 1.0 ± 0.1 | 4.5 ± 0.5** | 1.0 ± 0.1 | 4.2 ± 0.4*** |

The online version of this article includes the following source data for Table 2:

Source data 1. Early gene markers.

these $Trpm4$ cKO mice were compared with those in WT ($Trpm4^{+/+}$) mice. Haemodynamic and anatomical parameters obtained after 2 days and 14 days of sham/TAC in WT and $Trpm4$ cKO mice are shown in **Supplementary file 1**. TAC produced a similar degree of LV pressure overload in both WT (p<0.001) and $Trpm4$ cKO (p<0.001) mice when compared with sham-operated groups (**Figure 2A**) but did not alter heart rate (**Figure 2B**), cardiac contractility (**Figure 2C,D**), LW (**Figure 2E**), or BW (**Figure 2G**). **Figure 2F** illustrates representative images of WT and $Trpm4$ cKO mouse hearts after 14 days of sham or TAC. No LVH was detected 2 days after TAC in either $Trpm4$ cKO mice or WT mice when compared with sham-operated groups (**Figure 2H–J**). After 14 days, TAC induced a 32, 42, and 44% increase (all p<0.001) in HW/BW ratio, LVW/BW ratio, and LVW/TL ratio, respectively, in WT mice when compared with sham-operated controls (**Figure 2H–J**). However, this hypertrophic response to 14 days of TAC was attenuated in $Trpm4$ cKO mice, as evident by only a 17, 20, and 23% increase (all p<0.001) in HW/BW ratio, LVW/BW ratio, and LVW/TL ratio, respectively (**Figure 2H–J**). These findings demonstrate that when compared with WT mice, $Trpm4$ cKO mice developed approximately 50% less LVH (p<0.001) in response to TAC.

## Reduced fibrosis in $Trpm4$ cKO hearts after TAC

We evaluated cardiac fibrosis in response to pressure overload in $Trpm4$ cKO hearts and WT hearts by Masson's trichrome staining (**Figure 2K**). When compared with an average 3.17-fold increase (p<0.001) in cardiac fibrosis in WT TAC hearts, the increase in $Trpm4$ cKO TAC hearts was only 1.75-fold (p<0.05) (**Figure 2L**). In addition, we found a significant increase in collagen III ($Col3a1$) mRNA expression in WT TAC hearts when compared with WT sham hearts (p<0.001). However, there was no significant increase in collagen III ($Col3a1$) mRNA expression in $Trpm4$ cKO TAC hearts when compared with sham hearts (**Figure 2M**). Thus, $Trpm4$ inactivation attenuated the fibrotic response to TAC.

## TRPM4 deficiency reduced the expression of hypertrophy markers in response to TAC-induced pressure overload

Consistent with the development of pathological hypertrophy, both 2 and 14 days of TAC in WT mice significantly enhanced expression of the hypertrophy-associated genes, ANP ($Nppa$), BNP ($Nppb$), and α-SA ($Acta1$) (**Figure 3A,B**). However, these gene markers remained unchanged with TAC in $Trpm4$ cKO mice (**Figure 3A,B**), except for ANP ($Nppa$) at 14 days. These data indicate that loss of TRPM4 attenuates the activation of hypertrophic marker genes in response to TAC.

## CaMKII-HDAC4-MEF2 hypertrophic signalling pathway in WT and $Trpm4$ cKO mouse hearts

We next examined the molecular signalling pathways mediating LVH in both WT and $Trpm4$ cKO hearts after 2 days of TAC, a time at which molecular signalling is already activated in response to

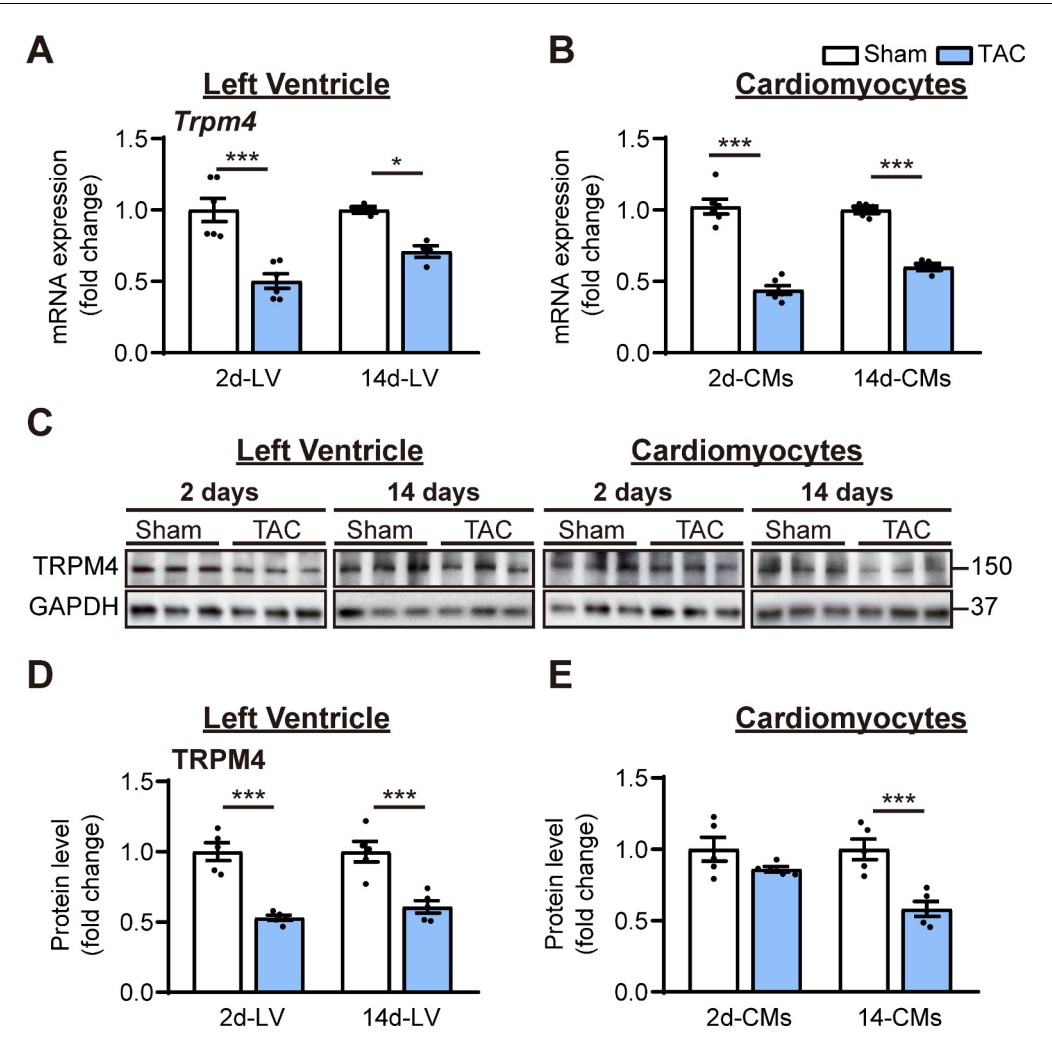

**Figure 1.** TRPM4 expression was downregulated in response to left ventricular (LV) pressure overload. (**A**) Relative mRNA expression of *Trpm4* in LV tissue and (**B**) in LV cardiomyocytes (CMs) after 2 days and 14 days of sham and TAC. (**C**) Representative western blots of TRPM4 protein expression in LV tissue (left panel) and in LV cardiomyocytes (right panel). (**D**) Western blots from LV tissue and (**E**) LV cardiomyocytes after 2 days and 14 days of TAC were quantified for TRPM4 protein expression. Relative TRPM4 mRNA and protein expression in the LV tissue and cardiomyocytes were normalised by GAPDH and calculated as fold change relative to sham in 2 days and 14 days groups, respectively. Results are presented as means ± SEM. *p<0.05, ***p<0.001 vs. sham-operated groups.

The online version of this article includes the following source data and figure supplement(s) for figure 1:

**Source data 1.** Source data file (Excel) for *Figure 1A,B,D and E*.

**Figure supplement 1.** Enlarged heart and cardiac fibrosis were detected 14 days after TAC.

the increased haemodynamic load induced by TAC (*Yu et al., 2021*), but before measurable LVH has developed.

The cytoplasmic and the nuclear fractions of LV tissue were separated as described in Materials and methods. High fraction purity was confirmed by western blot using antibodies against marker proteins specific for cytoplasmic (glyceraldehyde 3-phosphate dehydrogenase, GAPDH) and nuclear (Histone H2B) fractions (*Figure 4—figure supplement 1*).

Representative images of key cytoplasmic and nuclear proteins detected by western blot analysis are shown in *Figure 4A*. Quantitative data for cytoplasmic and nuclear proteins, normalised by GAPDH and Histone H2B, respectively, are shown in *Figure 4B*. We first investigated whether

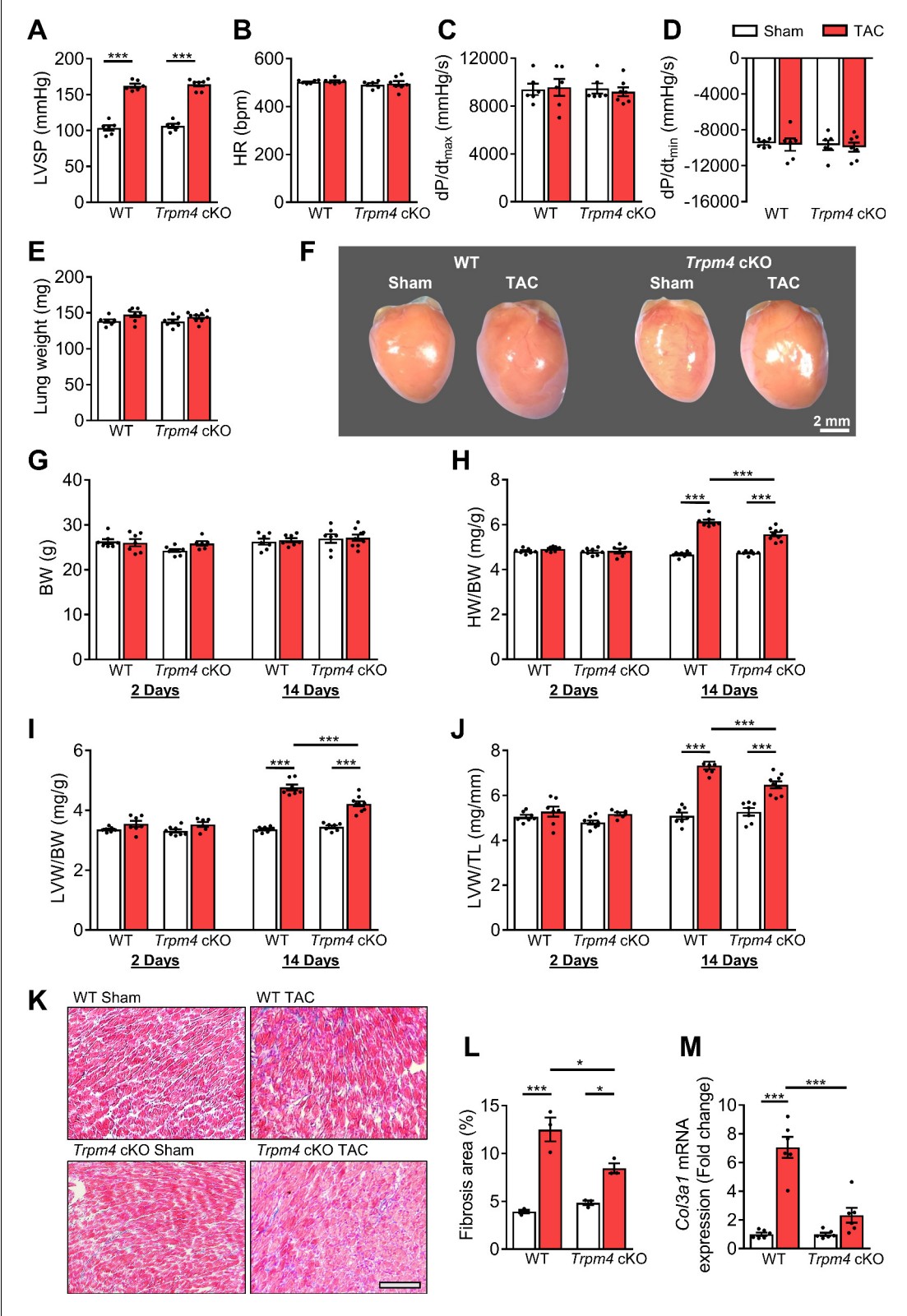

**Figure 2.** The hypertrophic response of WT and *Trpm4* cKO mice to TAC-induced LV pressure overload. (A) Systolic pressure, (B) heart rate, (C, D) dP/dt after 14 days of sham or TAC in WT and *Trpm4* cKO mice. (n = 6–7/group). (E) Lung weight after 14 days of sham or TAC in WT and *Trpm4* cKO mice. (n = 7–9/group). (F) Representative photos indicate heart size differences after 14 days of sham or TAC in WT and *Trpm4* cKO mice. (G) Body weight, (H) Heart weight, and (I, J) LV weight normalised to body weight and tibia length, in WT and *Trpm4* cKO mice after 2 days and 14 days of sham

*Figure 2 continued on next page*

*Figure 2 continued*

or TAC. (n = 7–9/group). (K) Representative micrographs and (L) quantitation of Masson's trichrome staining of LV tissue from WT mice and *Trpm4* cKO mice after 14 days of sham or TAC (n = 3/group), scale bar = 200 μm in (K). (M) Relative collagen III (*Col3a1*) mRNA expression after 14 days of sham or TAC. (n = 6/group). The mRNA relative expression was normalised by comparison to GAPDH and calculated as fold change relative to sham in WT and *Trpm4* cKO groups, respectively. Results are presented as means ± SEM. *p<0.05, ***p<0.001.

The online version of this article includes the following source data for figure 2:

**Source data 1.** Source data file (Excel) for *Figure 2A,B,C,D,E,G,H,I,J,L, and M*.

CaMKIIδ activity is increased in response to pressure overload by measuring both total CaMKIIδ expression and auto-phosphorylated CaMKII (p-CaMKII) at threonine 287 (Thr287), a marker of CaMKII activation (*Lai et al., 1987*; *Luczak et al., 2020*). We found that in WT hearts, 2 days of TAC resulted in a significant increase in the total CaMKIIδ protein level in cytoplasmic (p<0.01) and nuclear fractions (p<0.01), accompanied by increased p-CaMKII protein levels in both the cytoplasm (p<0.05) and nucleus (p<0.01). Associated with this increase, there was a rise in total cytoplasmic HDAC4 (p<0.01) and phosphorylated HDAC4 (p-HDAC4) levels (p<0.001), but no change in nuclear HDAC4. This 2.11-fold increase in the cytoplasmic/nuclear ratio of HDAC4 (p<0.01) in WT hearts indicates that TAC-induced pressure overload leads to the nuclear export of HDAC4 in WT TAC hearts. This increase was accompanied by a 1.76-fold increase of MEF2A levels in the nucleus (p<0.05), which together with the de-repression of MEF2A activity would account for the induction of LVH.

In contrast to the effects of TAC in WT hearts, in *Trpm4* cKO hearts, TAC produced a decrease in cytoplasmic CaMKIIδ levels (0.66-fold of that observed in sham hearts; p<0.001). Consistent with

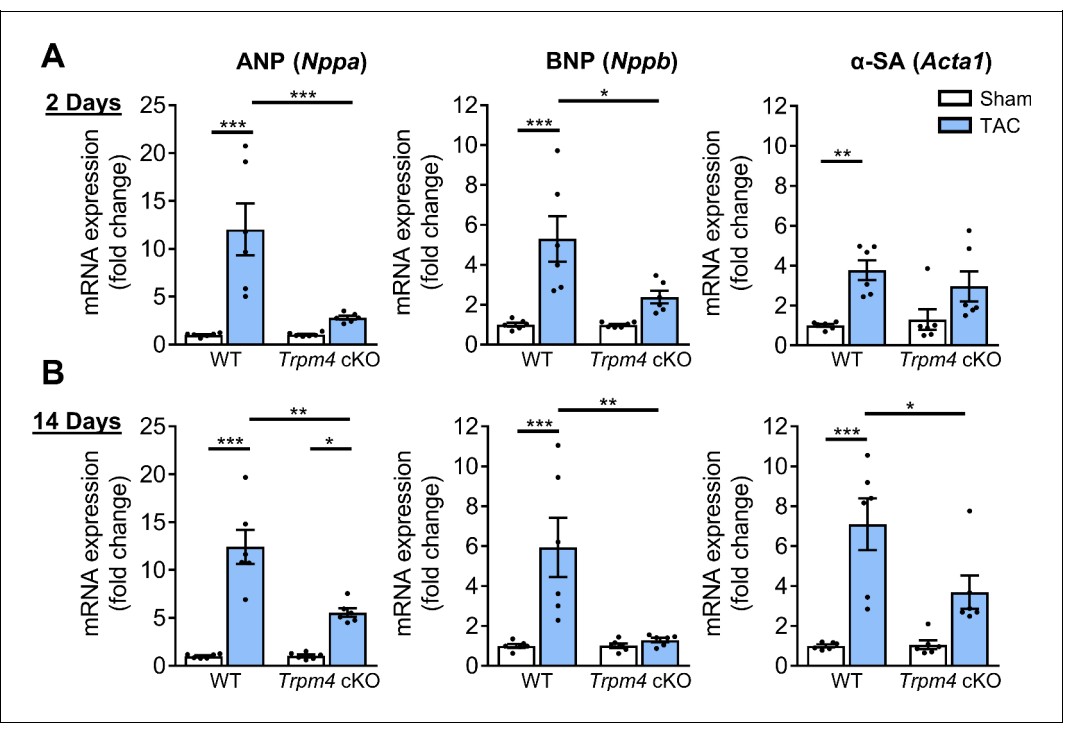

**Figure 3.** Comparison of gene expression of LVH markers in response to TAC-induced pressure overload in WT and *Trpm4* cKO mice. (A) Relative mRNA expression of ANP (*Nppa*), BNP (*Nppb*), and α-SA (*Acta1*) after 2 days of TAC compared to sham-operated mice. (n = 6/group). (B) Relative mRNA expression of ANP (*Nppa*), BNP (*Nppb*), and α-SA (*Acta1*) after 14 days of sham and TAC. (n = 6/group). The mRNA relative expression was normalised by GAPDH and calculated as fold change relative to WT sham in 2 days and 14 days groups, respectively. Results are presented as means ± SEM, *p<0.05, **p<0.01, ***p<0.001.

The online version of this article includes the following source data for figure 3:

**Source data 1.** Source data file (Excel) for *Figure 3A,B*.

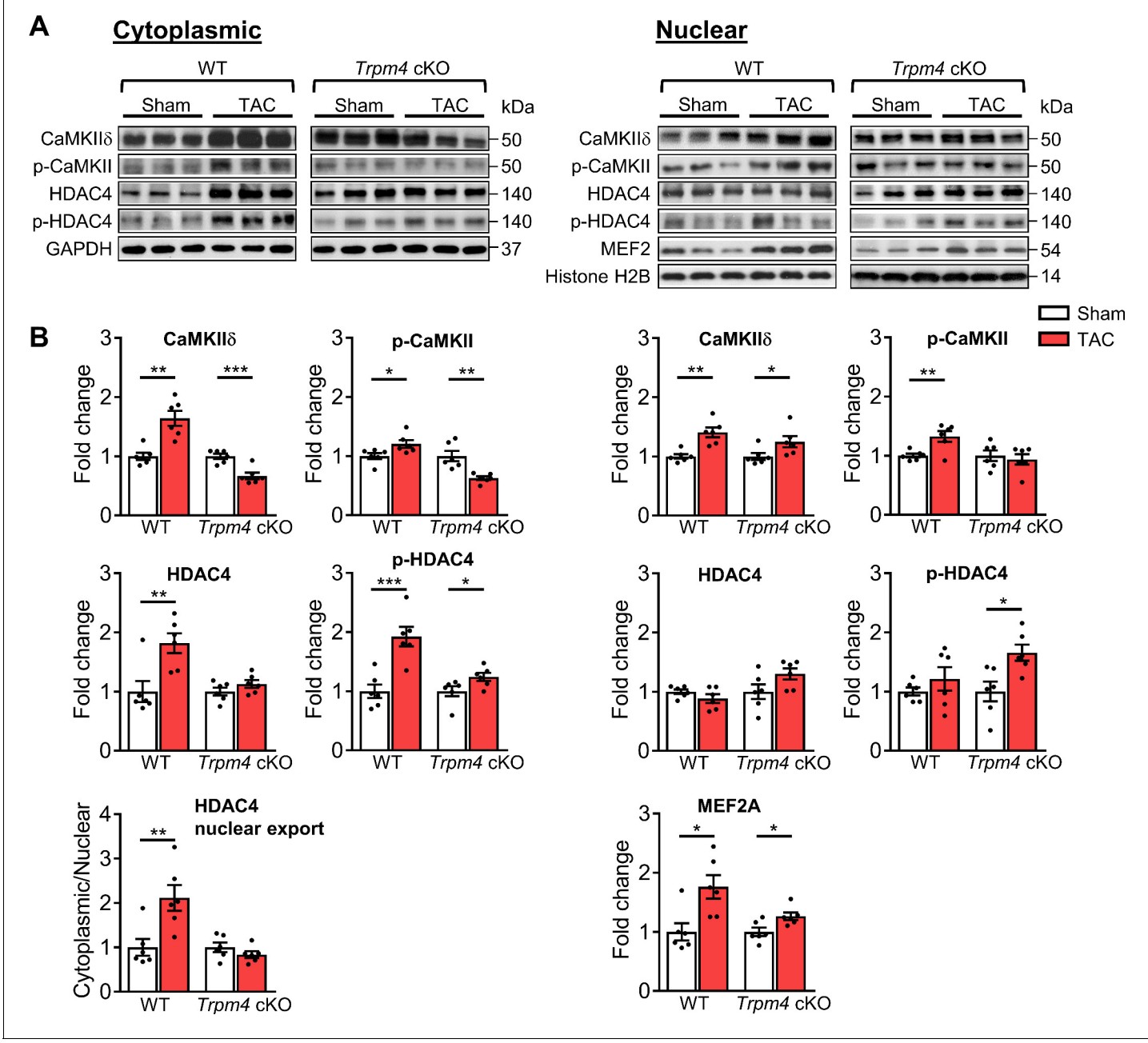

**Figure 4.** CaMKII-HDAC4-MEF2 signalling pathway in response to TAC after 2 days in WT and *Trpm4* cKO mouse hearts. (**A**) Representative western blots showing the expression of key proteins in the CaMKII-HDAC4-MEF2 signalling pathway in the cytoplasm (left) and nucleus (right). (**B**) Cytoplasmic (left) and nuclear (right) quantitative data were normalised by GAPDH and Histone H2B, respectively. Fold changes and cytoplasmic/nuclear ratios were calculated relative to sham groups, in each genotype. Results are presented as means ± SEM, n = 6/group, *p<0.05, **p<0.01, ***p<0.001.
The online version of this article includes the following source data and figure supplement(s) for figure 4:

**Source data 1.** Source data file (Excel) for *Figure 4B*.

**Figure supplement 1.** Demonstration of successful fractionation of compartments.

**Figure supplement 2.** Cytoplasmic/nuclear ratio of the key proteins in CaMKII-HDAC4-MEF2 signalling pathway.

**Figure supplement 2—source data 1.** Source data file (Excel) for *Figure 4—figure supplement 2*.

this, the cytoplasmic p-CaMKII levels in TAC hearts also decreased (0.63-fold of that observed in sham hearts; p<0.01). Although the increase in nuclear CaMKIIδ was similar (p<0.05) to that observed with TAC in WT hearts, the nuclear p-CaMKII levels showed no significant difference between sham and TAC in *Trpm4* cKO hearts, which indicates less nuclear CaMKIIδ activation by TAC in *Trpm4* cKO hearts. We found that p-HDAC4 increased in *Trpm4* cKO TAC hearts in both the cytoplasm (p<0.05) and the nucleus (p<0.05), but there was no change in total HDAC4. Thus, the cytoplasmic/nuclear ratio of HDAC4 remained the same in *Trpm4* cKO TAC hearts as in sham hearts, indicating inhibition of nuclear HDAC4 export in TAC-treated *Trpm4* cKO hearts. In addition, consistent with MEF2A activation driving hypertrophy development, reduced LVH in *Trpm4* cKO TAC hearts was associated with a smaller (1.26-fold) increase in MEF2A levels in the nucleus (p<0.05) when compared with WT TAC hearts. For completeness, the cytoplasmic/nuclear ratios of other relevant proteins in this signalling pathway are shown in *Figure 4—figure supplement 2*.

Taken together, these data implicate the CaMKII-HDAC4-MEF2 hypertrophic signalling pathway in mediating TAC-induced LVH, but the extent of the hypertrophic response is regulated by TRPM4 channels.

## Calcineurin-NFAT hypertrophic signalling pathway in WT and *Trpm4* cKO mouse hearts

Next, we examined the expression of proteins involved in the calcineurin-NFAT hypertrophic signalling pathway. Representative images of key cytoplasmic and nuclear proteins detected by western blot analysis are shown in *Figure 5A*. Quantitative data for cytoplasmic and nuclear proteins, normalised by GAPDH and Histone H2B, respectively, are shown in *Figure 5B*. In WT hearts, there was no significant difference in cytoplasmic or nuclear NFATc4 protein expression in sham and TAC hearts after 2 days. Consistent with these findings, total GSK3β, serine-9 phosphorylated GSK3β and GATA4 levels were also unchanged in response to TAC. These findings indicate that TAC did not result in increased nuclear translocation of NFATc4, the most reliable indicator of calcineurin activation (*Molkentin, 2013*), and confirm our previous finding (*Yu et al., 2021*) that the calcineurin-NFAT pathway is not activated by TAC.

In contrast to WT hearts, a 1.28-fold increase in nuclear NFATc4 (p<0.05) was observed in *Trpm4* cKO hearts after TAC, which led to a 0.31-fold decrease in the cytoplasmic/nuclear ratio compared to sham-operated hearts (p<0.01). This indicated lower nuclear export of NFATc4 in the *Trpm4* cKO TAC hearts when compared to sham hearts. Accordingly, we found a 1.20-fold increase in nuclear p-GSK3β (serine 9, Ser9) (p<0.05) in *Trpm4* cKO TAC hearts. As phosphorylation at the serine 9 residue indicates inactivation of GSK3β, these findings suggest that the GSK3β-mediated export of NFATc4 from the nucleus was partially inhibited, which is consistent with the increased level of NFATc4 in the nucleus. Furthermore, accompanied by the increase in nuclear NFATc4, a 1.18-fold increase in GATA4 expression (p<0.05) in the nucleus was observed in *Trpm4* cKO TAC hearts. All these observations are consistent with a reduction in the tonic inhibition of calcineurin by CaMKII (*De Koninck and Schulman, 1998*; *Kreusser et al., 2014*; *MacDonnell et al., 2009*) in *Trpm4* cKO hearts after TAC. For completeness, the cytoplasmic/nuclear ratios of other relevant proteins in this signalling pathway are shown in *Figure 5—figure supplement 1*.

## Discussion

In the present study, we employed mice subjected to TAC as an in vivo cardiac hypertrophy model to investigate the role of the TRPM4 ion channels in pressure overload-induced pathological LVH. We compared *Trpm4* cKO mice with WT controls. The experimental animals were examined 2 days after surgery when the molecular signalling pathway that drives LVH is switched on in response to the increased haemodynamic load induced by TAC but, importantly, before LVH has developed. In addition, the experimental animals were examined 14 days after surgery when the TAC-induced LVH phenotype is evident.

First, we found that TRPM4 channel expression in the WT mouse heart was modified by TAC-induced pressure overload hypertrophy. At 2 days and 14 days after TAC, both *Trpm4* mRNA and protein expression were downregulated in LV tissue and isolated cardiomyocytes, suggesting that TRPM4 plays a role in TAC-induced LVH. Second, we demonstrated that the role of TRPM4 was pro-hypertrophic by performing sham and TAC surgery in *Trpm4* cKO mice. This demonstrated that a

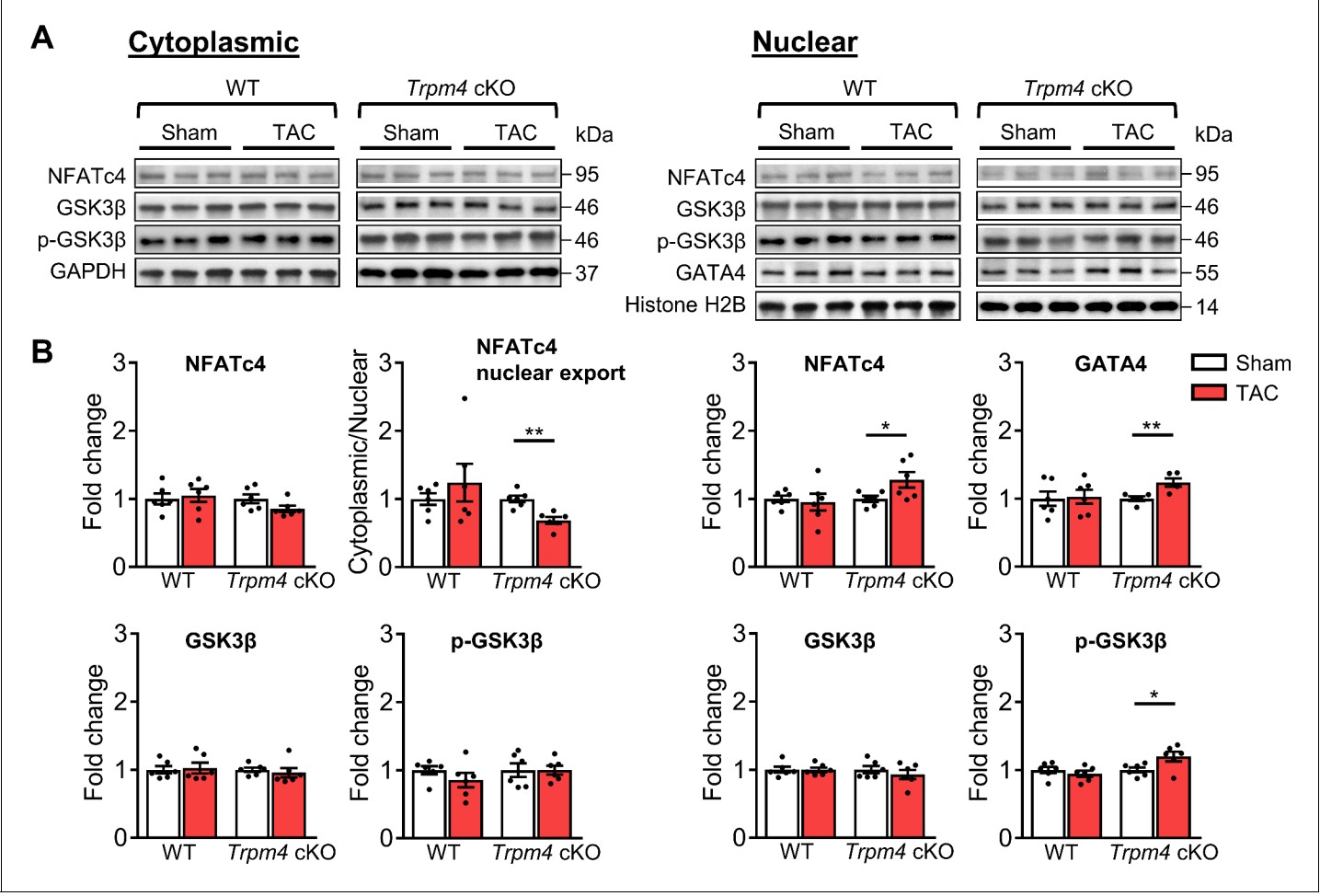

**Figure 5.** Calcineurin-NFAT signalling pathway in response to TAC after 2 days in WT and *Trpm4* cKO mouse hearts. (**A**) Representative western blots showing the expression of key proteins in the calcineurin-NFAT signalling pathway in cytoplasm (left) and nucleus (right). (**B**) Cytoplasmic (left) and nuclear (right) quantitative data were normalised by GAPDH and Histone H2B, respectively. Fold changes and cytoplasmic/nuclear ratios were calculated relative to sham groups, in each genotype. Results are presented as means ± SEM, n = 5–6/group, *p<0.05, **p<0.01.
The online version of this article includes the following source data and figure supplement(s) for figure 5:

**Source data 1.** Source data file (Excel) for *Figure 5B*.

**Figure supplement 1.** Cytoplasmic/nuclear ratio of the key proteins in calcineurin-NFAT signalling pathway.

**Figure supplement 1—source data 1.** Source data file (Excel) for *Figure 5—figure supplement 1*.

reduction in TRPM4 expression in cardiomyocytes dampens the hypertrophic response to TAC, as evident by an approximately 50% reduction in the degree of LVH and LV fibrosis in *Trpm4* cKO animals at 14 days after TAC, as compared with WT animals. Finally, to investigate the hypertrophic signalling pathways activated in response to pressure overload, we examined both the CaMKII-HDAC4-MEF2 and calcineurin-NFAT signalling pathways 2 days after TAC in WT and *Trpm4* cKO mice (*Figure 6*). This confirmed our previous finding that the CaMKII-HDAC4-MEF2 pathway was activated in response to TAC in WT mice (*Yu et al., 2021*), but also revealed the new finding of reduced activation of the CaMKII-HDAC4-MEF2 pathway after TAC in *Trpm4* cKO animals.

Previous studies reported that the TRPM4 current contributes to the mammalian atrial action potential (*Simard et al., 2013*) as well as to the notch and early repolarisation phases of the action potential in Purkinje cells (*Hof et al., 2016*), providing a potential link to cardiac arrhythmias (*Guinamard et al., 2015*; *Hedon et al., 2021*; *Wang et al., 2018*). Importantly, there is evidence that the TRPM4 channel is a critical modulator of ventricular remodelling in cardiac hypertrophy and heart failure (*Frede et al., 2020*; *Jacobs et al., 2015*; *Kecskés et al., 2015*; *Mathar et al., 2014*).

*Kecskés et al., 2015* reported that TRPM4 activation suppresses angiotensin II-induced cardiac hypertrophy, which is dependent on the activation of the calcineurin-NFAT pathway. It has been proposed that this is due to the $Ca^{2+}$-dependent modulation of TRPM4 activity, which leads to membrane depolarisation in cardiomyocytes and thus reduces the driving force for $Ca^{2+}$ influx via store-operated calcium entry (SOCE) through TRP canonical type 1 (TRPC1) and type 3 (TRPC3) ion channels (*Kecskés et al., 2015*; *Wu et al., 2004*).

To our knowledge, however, a role for TRPM4 in the LVH induced by mechanical pressure overload has not been demonstrated previously. We propose here that a mechanical stimulus, such as that exerted by TAC, is converted to downstream $Ca^{2+}$ signalling via the activity of mechanosensitive ion channels in the plasma membrane. Although the mechanosensitivity of TRP-type ion channels is still the subject of debate (*Cox et al., 2019*; *Gottlieb et al., 2008*), mammalian TRP ion channels, including TRPM4, have recently been shown to be insensitive to membrane stretch (*Constantine et al., 2016*; *Nikolaev et al., 2019*). Therefore, TRPM4 does not appear to be the primary mechanosensor responding to pressure overload. It is more likely to be a secondary ionotropic receptor downstream of a $Ca^{2+}$-permeable mechanosensitive ion channel, such as Piezo1 (*Gnanasambandam et al., 2015*; *Syeda et al., 2016*) or TRPV2/4 (*Lieben and Carmeliet, 2012*), that functions as the primary mechanoreceptor responding directly to pressure overload and thus initiating the hypertrophic response in TAC, which is not dependent on activation of the calcineurin-NFAT pathway (*Yu et al., 2021*).

Stimulated by the local $Ca^{2+}$ influx through a $Ca^{2+}$-permeable mechanosensitive ion channel, the $Na^+$-permeable TRPM4 activity then could either induce reverse activity of the $Na^+/Ca^{2+}$ exchanger through local $Na^+$ loading (*Conway and Koushik, 2001*; *Wang et al., 2018*) or depolarise the cardiomyocyte cell membrane to stimulate voltage-gated $Ca^{2+}$ channels. Such potential downstream ion channels include the L-type $Ca^{2+}$ channels, which were reported to mediate hypertrophic cardiomyopathy (*Viola and Hool, 2017*), as well as the T-type $Ca^{2+}$ channels whose splice variants were found to be regulated in rat LV hypertrophic hearts induced by aortic constriction (*Cribbs, 2010*). Either of these outcomes would lead to a high-amplitude increase in local $Ca^{2+}$. Thus, as a $Ca^{2+}$-dependent non-selective monovalent cation channel (*Constantine et al., 2016*; *Launay et al., 2002*; *Nilius et al., 2003*; *Nilius et al., 2005*), TRPM4 could contribute to TAC-induced LVH by modulating downstream voltage-gated $Ca^{2+}$ ion channels or the $Na^+/Ca^{2+}$ exchanger (*Figure 6*).

In this study, we confirmed the involvement of TRPM4 in TAC-induced LVH using *Trpm4* cKO mice. Despite identical TAC-induced increases in haemodynamic load in both WT and *Trpm4* cKO mice, the latter displayed a significantly reduced LVH response. This is in contrast to the increased hypertrophy reported in angiotensin II-treated *Trpm4* cKO mice that is mediated by the calcineurin-NFAT pathway (*Kecskés et al., 2015*). These differential effects of TRPM4 on angiotensin II-mediated (*Kecskés et al., 2015*) and TAC-induced LVH support our previous finding that these two hypertrophic stimuli are mediated by distinct signalling mechanisms. Thus, in agreement with the present findings, we showed previously that the CaMKII-HDAC4-MEF2 pathway, but not the calcineurin-NFAT signalling pathway, is activated in response to TAC-induced pressure overload (*Yu et al., 2021*). This is most likely because CaMKIIδ and calcineurin respond to different characteristics of intracellular $Ca^{2+}$ signalling (*De Koninck and Schulman, 1998*; *Dolmetsch et al., 1997*). Whereas calcineurin activation requires a sustained increase in the resting intracellular $Ca^{2+}$ concentration, CaMKIIδ activation is more sensitive to high-frequency/high amplitude $Ca^{2+}$ oscillations (*Colella et al., 2008*; *De Koninck and Schulman, 1998*), which are known to occur with TAC-induced aortic constriction (*Chen et al., 2011*). CaM responds to this high-amplitude $Ca^{2+}$ stimulus through the lower affinity $Ca^{2+}$ binding site at its N-lobe (*Evans and Shea, 2009*; *Saimi and Kung, 2002*), which subsequently activates CaMKIIδ and thus stimulates the CaMKII-HDAC4-MEF2 pathway (*Yu et al., 2021*). Moreover, once activated, CaMKII has been shown to inhibit calcineurin activity (*Kreusser et al., 2014*; *MacDonnell et al., 2009*; *Figure 6*).

The high-frequency/high-amplitude $Ca^{2+}$ oscillations characteristic of TAC and the consequent intermittent nature of the resulting TRPM4 activation, as distinct from the persistent activation associated with angiotensin II-induced hypertrophy, may mean that TAC-induced TRPM4 activation does not reduce the driving force for $Ca^{2+}$ influx via SOCE that characterises the anti-hypertrophic action of TRPM4 in angiotensin II-induced hypertrophy (see above). Rather, in TAC-induced hypertrophy, TRPM4 acts as a second messenger, amplifying the small load-dependent $Ca^{2+}$ signal produced by a mechanosensitive $Ca^{2+}$-permeable ion channel. This would account for the pro-hypertrophic effect

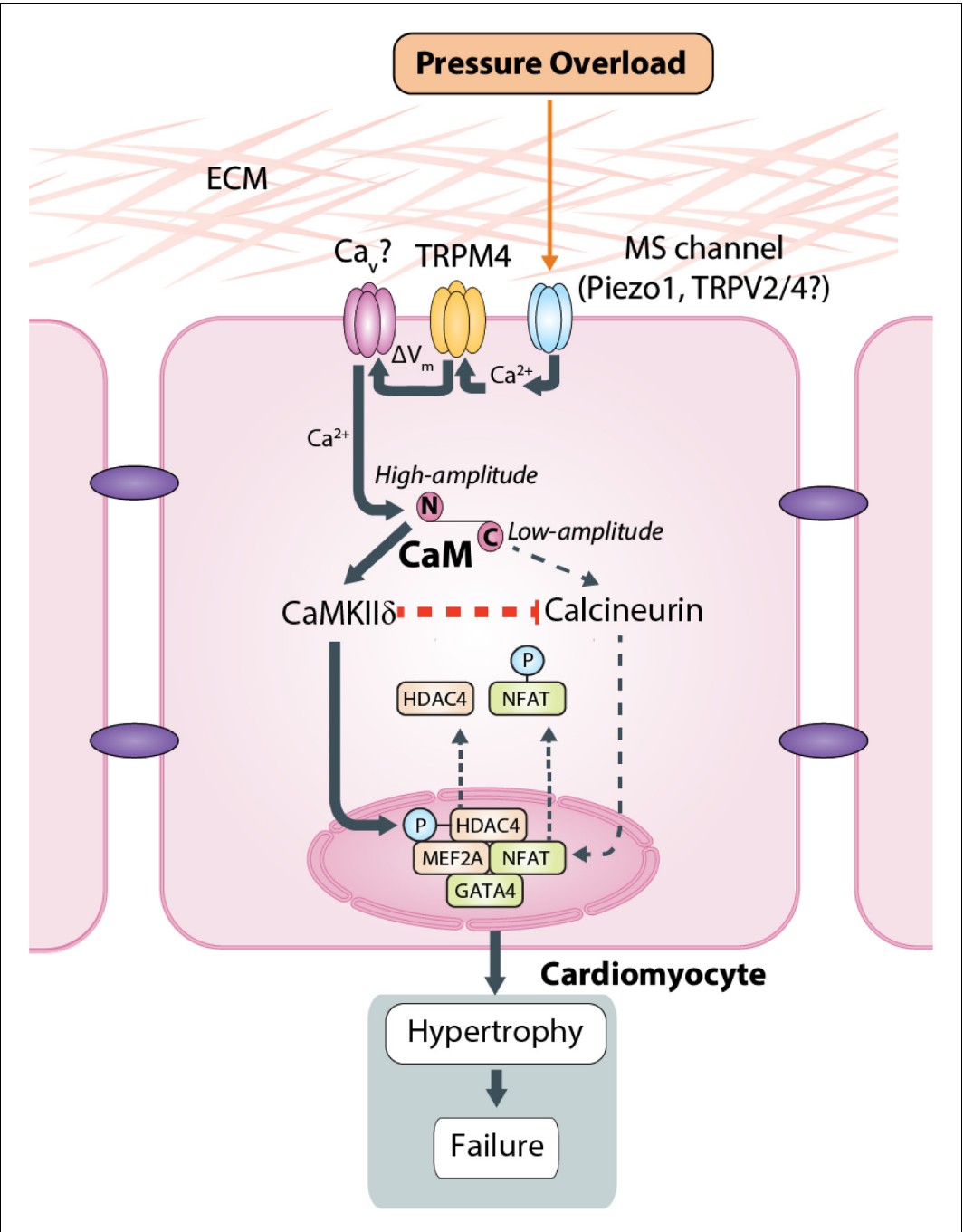

**Figure 6.** Schematic of the putative TAC-induced pathway that culminates in left ventricular hypertrophy. A $Ca^{2+}$-permeable MS channel (e.g. Piezo1, TRPV2, TRPV4) acts as the mechanotransducer providing local $Ca^{2+}$ that in turn stimulates TRPM4. The $Na^+$-permeable TRPM4 activity then could either stimulate voltage-gated $Ca^{2+}$ channels through membrane depolarisation or induce reverse activity of the $Na^+$/ $Ca^{2+}$ exchanger through local $Na^+$ loading. Either of these outcomes would lead to a high-amplitude increase in local $Ca^{2+}$. Calmodulin then responds to this high-amplitude $Ca^{2+}$ stimulus through the lower affinity $Ca^{2+}$ binding site at its N-lobe which subsequently activates CaMKIIδ and thus stimulates the CaMKII-HDAC4-MEF2 pathway as shown in *Yu et al., 2021*. Calcineurin activation is inhibited by the activated CaMKIIδ and is activated preferentially by low-amplitude $Ca^{2+}$ signalling via Gq-coupled receptors and calmodulin (see Discussion). ECM: extracellular matrix, MS: mechanosensitive, $Ca_v$: voltage-gated $Ca^{2+}$ channel, $\Delta V_m$: membrane depolarisation, CaM: calmodulin.

of TRPM4 in TAC-induced hypertrophy as distinct from its anti-hypertrophic effect in angiotensin II-induced hypertrophy.

In terms of the signalling pathway mediating pressure overload-induced LVH, we found that in *Trpm4* cKO TAC hearts, the reduced LVH response was associated with significantly less activation of the CaMKII-HDAC4-MEF2 pathway, with reduced CaMKIIδ activation resulting in reduced nuclear export of HDAC4. Since nuclear HDAC4 inhibits MEF2A activity, a reduction in HDAC4 nuclear export would result in diminished MEF2A disinhibition and, given that MEF2A is a critical nuclear transcriptional regulator causing pathological cardiac remodelling, reduced hypertrophy development (*Passier et al., 2000*) as, indeed, observed here in *Trpm4* cKO TAC hearts.

Decreased expression of TRPM4 channels in *Trpm4* cKO animals likely modifies $Ca^{2+}$-signalling, which directly regulates CaMKIIδ activation and its downstream pathway in response to TAC. Comparable with a study reporting that blockade of MEF2 acetylation can permit recovery from pathological cardiac hypertrophy without impairing physiologic adaptation (*Wei et al., 2017*), the lower concentration and reduced activity of MEF2A that we found in *Trpm4* cKO TAC hearts suggest that inhibition of TRPM4 channels is potentially a viable therapeutic option for reducing pathological hypertrophy in response to pressure overload.

Interestingly, although the calcineurin-NFAT hypertrophic signalling pathway is not activated by TAC in WT hearts, it was partially activated in *Trpm4* cKO TAC hearts, which manifested itself in the inhibition of GSK3β-mediated NFATc4 nuclear export and by an increase in GATA4. This may be explained by the reduction of the cytoplasmic CaMKIIδ in *Trpm4* cKO TAC hearts, as CaMKII negatively regulates calcineurin activity (*Kreusser et al., 2014*; *MacDonnell et al., 2009*; *Figure 6*). It is notable, nevertheless, that the net effect of the loss of TRPM4 was a significant reduction in TAC-induced LVH, indicating that the direct effect of less activation of the CaMKII-HDAC4-MEF2 pathway in reducing hypertrophy development outweighed the indirect pro-hypertrophic effect resulting from blunting CaMKIIδ's inhibition of calcineurin.

In summary, our study provides compelling evidence that TRPM4 plays an important role in pressure overload-induced pathological LVH, with diminished TRPM4 expression reducing TAC-induced hypertrophy. Furthermore, we demonstrated that TRPM4 is a likely component of a cardiac mechanotransduction process that activates the CaMKII-HDAC4-MEF2 pathway in response to TAC. It is likely that TRPM4 is activated by upstream primary mechanoreceptors, such as Piezo1 or TRPV2/4 channels, which provide the first step in this mechanotransduction pathway. These findings expand our understanding of the molecular mechanism underlying mechanical pressure overload-induced LVH. Moreover, our work provides new insights into possible treatment strategies for limiting pressure overload-induced pathological hypertrophy.

## Methods

### Key resources table

| Reagent type (species) or resource | Designation | Source or reference | Identifiers | Additional information |
|---|---|---|---|---|
| Antibody | Anti-TRPM4 (rabbit polyclonal) | Alomone Labs | Cat# ACC-044, RRID:AB_2040250 | Western blot (1:200) |
| Antibody | Anti-CaMKII delta (rabbit monoclonal) | Abcam | Cat# ab181052, RRID:AB_2891241 | Western blot (1:1000) |
| Antibody | Anti-p-CaMKII (Thr287) (rabbit polyclonal) | Thermo Fisher Scientific | Cat# PA5-37833, RRID:AB_2554441 | Western blot (1:5000) |
| Antibody | Anti-HDAC4 (rabbit monoclonal) | Cell Signaling Technology | Cat# 7628 RRID:AB_10860255 | Western blot (1:1500) |
| Antibody | Anti-p-HDAC4 (Ser246) (rabbit monoclonal) | Cell Signaling Technology | Cat# 3443 RRID:AB_2118723 | Western blot (1:1500) |
| Antibody | Anti-MEF2A (rabbit polyclonal) | Cell Signaling Technology | Cat# 9736 RRID:AB_10691852 | Western blot (1:3000) |
| Antibody | Anti-NFATc4 (rabbit polyclonal) | Abcam | Cat# ab99431, RRID:AB_10675673 | Western blot (1:1500) |

*Continued on next page*

*Continued*

| Reagent type (species) or resource | Designation | Source or reference | Identifiers | Additional information |
|---|---|---|---|---|
| Antibody | Anti-GSK3β (rabbit monoclonal) | Cell Signaling Technology | Cat# 9315, RRID:AB_490890 | Western blot (1:500) |
| Antibody | Anti-p-GSK3β (Ser9) (rabbit polyclonal) | Cell Signaling Technology | Cat# 9336, RRID:AB_331405 | Western blot (1:1500) |
| Antibody | Anti-GATA4 (mouse monoclonal) | Santa Cruz Biotechnology | Cat# sc-25310, RRID:AB_627667 | Western blot (1:1000) |
| Antibody | Anti-GAPDH (rabbit monoclonal) | Cell Signaling Technology | Cat# 2118, RRID:AB_561053 | Western blot (1:10,000) |
| Antibody | Anti-Histone H2B (rabbit polyclonal) | Abcam | Cat# ab1790, RRID:AB_302612 | Western blot (1:5000) |
| Antibody | Goat anti-rabbit IgG (goat polyclonal) | Abcam | Cat# ab6721, RRID:AB_955447 | Western blot (1:10,000) |
| Antibody | Rabbit anti-mouse IgG (rabbit polyclonal) | Abcam | Cat# ab6728, RRID:AB_955440 | Western blot (1:5000) |
| Sequence-based reagent | ANP (*Nppa*)_F | Sigma-Aldrich | PCR primers | TGATAGATGAAGGCAGGAAGCCGC |
| Sequence-based reagent | ANP(*Nppa*)_R | Sigma-Aldrich | PCR primers | AGGATTGGAGCCCAGAGTGGACTAGG |
| Sequence-based reagent | BNP (*Nppb*)_F | Sigma-Aldrich | PCR primers | TCTCCAGAGCAATTCAAGAT |
| Sequence-based reagent | BNP (*Nppb*)_R | Sigma-Aldrich | PCR primers | AACAACTTCAGTGCGTTACA |
| Sequence-based reagent | α-SA (*Acta1*)_F | Sigma-Aldrich | PCR primers | GTGAGATTGTGCGCGACATC |
| Sequence-based reagent | α-SA (*Acta1*)_R | Sigma-Aldrich | PCR primers | GGCAACGGAAACGCTCATT |
| Sequence-based reagent | Collagen III (*Col3A1*)_F | Sigma-Aldrich | PCR primers | GACAGATTCTGGTGCAGAGA |
| Sequence-based reagent | Collagen III (*Col3A1*)_R | Sigma-Aldrich | PCR primers | CATCAACGACATCTTCAGGAAT |
| Sequence-based reagent | *Trpm4*_F | Sigma-Aldrich | PCR primers | GAGAAGCCCACAGATGCCTATG |
| Sequence-based reagent | *Trpm4*_R | Sigma-Aldrich | PCR primers | AGCACCGACACCACCAAGTTTG |

## Mice

In the first part of the study, we performed experiments on 11–13 week old male C57BL/6J WT mice at the Victor Chang Cardiac Research Institute, Australia. In the second part of this study, we performed surgery on C57BL/6N WT and age- and sex-matched cardiac-specific *Trpm4* cKO mice in Katholieke Universiteit Leuven, Belgium. *Trpm4*^flox mice were crossbred with *Myl7*-Cre mice to generate the *Trpm4* cKO mice (*Kecskés et al., 2015*). All animals were entered into the study in a randomised order, and the investigators were blinded to genotype. All experimental procedures were approved by the Animal Ethics Committee of Garvan/St Vincent's (Australia) or Katholieke Universiteit Leuven (Belgium), respectively, in accordance with the guidelines of both the Australian code for the care and use of animals for scientific purposes (8th edition, National Health and Medical Research Council, AU, 2013) and the Guide for the Care and Use of Laboratory Animals (8th edition, National Research Council, USA, 2011).

## Induction of LVH

WT and *Trpm4* cKO mice were subjected to TAC to induce pressure overload. Mice were anesthetised with 5% isoflurane and ventilated at 120 breaths/min (Harvard Apparatus Rodent Ventilator). The transverse aortic arch was accessed via an incision in the second intercostal space and constricted with a ligature tied around a 25-gauge needle, which was then removed. The TAC procedure was modified from a published paper (*Rockman et al., 1994*). Sham mice underwent the same procedure but the ligature was not tied. Simultaneous direct pressure recordings (1.4 F pressure catheter, AD Instruments, P/L) from both the right carotid artery and the aorta distal to the ligature (n = 20 mice) indicated a TAC pressure gradient of 60 ± 8 mmHg with this technique. Animals were sacrificed after 2 days or 14 days.

## Invasive haemodynamic measurements

After 14 days of sham or TAC, mice were anesthetised by inhalation of isoflurane (1.5%) and a 1.4 F micro-tip pressure catheter (Millar Instruments Inc, Houston, TX) was inserted into the left ventricle via the right carotid artery. The heart rate, systolic aortic pressure, LV systolic pressure, +dP/dt, and −dP/dt were recorded (LabChart 6 Reader, AD Instruments, P/L). Animals were sacrificed, and the heart weight (HW) and left ventricle weight (LVW) normalised to BW and to tibia length (TL) were measured as indicators of LVH.

## Mouse LV cardiomyocytes isolation and purification

WT mice were heparinised and euthanised according to the Animal Research Act 1985 No 123 (New South Wales, Australia). Hearts were dissected and perfused through the aorta and the coronary arteries by 10 ml pH 7.2 perfusion buffer containing 135 mM NaCl, 4 mM KCl, 1 mM MgCl$_2$, 0.33 mM NaH$_2$PO$_2$, 10 mM HEPES, 10 mM glucose, 10 mM 2,3-butanedione 2-monoxime (BDM), and 5 mM taurine, with a Langendorff apparatus at 37°C for 5 min. Next, 30 ml digestion buffer composed of the above solution and Collagenase B, D (dose by BW: 0.4 mg/g, Roche) and Protease Enzyme Type XIV (dose by BW: 0.07 mg/g, Sigma-Aldrich) was used to perfuse the hearts for 15 min. After the perfusion, the heart was removed from the setup and placed into a pH 7.4 transfer buffer containing 135 mM NaCl, 4 mM KCl, 1 mM MgCl$_2$, 0.33 mM NaH$_2$PO$_2$, 10 mM HEPES, 5.5 mM glucose, 10 mM BDM, and 5 mg/ml BSA. Both atria and the right ventricle were discarded, and the LV muscle was torn into small pieces and gently dispensed into the transfer buffer repeatedly with a pipette to isolate cardiomyocytes. The suspension was then filtered through a 200 micro filcon cup filter (BD) and centrifuged at 20 g for 2 min. After that, the cardiomyocytes were purified by a method described in a separate paper (Nicks et al., 2020). We confirmed that rod-shaped cardiomyocytes accounted for more than 85% of the total purified cardiomyocytes. The isolated cardiomyocytes were frozen immediately in liquid nitrogen and stored at −80°C for following experiments.

## Quantitative real-time polymerase chain reaction

Gene expression was determined by quantitative RT-PCR. Total RNA was extracted and purified from LV tissue and isolated cardiomyocytes with the RNeasy Fibrous Tissue Mini Kit (QIAGEN), following the manufacturer's protocol. RNA (500 ng) was reverse transcribed into cDNA using the SuperScript III First-Strand Synthesis SuperMix kit (Invitrogen). cDNA was subjected to PCR amplification to detect ANP (*Nppa*), BNP (*Nppb*), α-SA (*Acta1*), collagen III (*Col3a1*), and *Trpm4* gene expression, performed with the CFX384 Touch Real-Time PCR Detection System (Bio-Rad), PCR master mix LightCycler 480 SYBR Green I Master (Invitrogen). Samples were run in technical triplicate, and the mRNA expression levels were normalised to those of GAPDH to calculate relative gene expression using delta-delta Ct method. The mouse RT-PCR primers (Sigma-Aldrich) used are shown in (Key resources table).

## Western blotting

For total protein extraction, LV tissue and isolated cardiomyocytes were lysed in a pH 7.4 lysis buffer containing 150 mM NaCl, 50 mM Tris–HCL, 1% Triton X-100, 1 mM sodium orthovanadate, 1 mM beta-glycerophosphate, 5 mM dithiothreitol, and MiniComplete protease inhibitors (Roche); for cytoplasmic and nuclear protein extraction, LV tissue was lysed using NE-PER nuclear and cytoplasmic extraction reagents (Pierce Biotechnology) and Protesase Inhibitor Cocktail Kit and Halt Phosphatase Inhibitor Cocktail (Pierce Biotechnology), both with a homogeniser (PRO Scientific). Protein (40 μg for each sample) was loaded on 4–20% Mini-PROTEAN TGX Gels (Bio-Rad) and separated by electrophoresis. Samples were transferred to PVDF membranes (Bio-Rad), blocked with 5% bovine serum albumin (BSA), and then labelled overnight with primary antibodies (Key resources table): anti-TRPM4 (1:200, Alomone Labs), anti-CaMKIIδ (1:1000; Abcam), anti-p-CaMKII (Thr287, 1:5000; Thermo Scientific), anti-HDAC4 (1:1500; Cell Signaling Technology), anti-p-HDAC4 (Ser246, 1:1500; Cell Signalling Technology), anti-MEF2A (1:3000; Cell Signalling Technology), anti-NFATc4 (1:1500; Abcam), anti-GSK3β (1:500; Cell Signaling Technology), anti-p-GSK3β (Ser9, 1:1500; Cell Signaling Technology), and anti-GATA4 (1:1000; Santa Cruz Biotechnology). Anti-GAPDH (1:10000; Cell Signaling Technology) and anti-Histone H2B (1:5000; Abcam) were used to standardise for loading. Horseradish peroxidase-conjugated goat anti-rabbit (1:10,000) or rabbit anti-mouse (1:5000)

secondary antibodies (Abcam) (Key resources table) were used at room temperature for 1 h. Immunologic detection was accomplished using Amersham ECL Western blotting detection reagents (GE Healthcare). Protein levels were quantified by densitometry using ImageJ (NIH) software. Protein levels were normalised to relative changes in Histone H2B for the nuclear fraction and GAPDH for the cytoplasmic fraction and expressed as fold changes relative to those of control animals.

## Histology

Masson's trichrome stain was used to quantify fibrosis in the LV (collagen fibres stain blue). The hearts were excised from isoflurane-euthanised mice and washed with phosphate-buffered saline. Then the hearts were longitudinally cut at the frontal plane, embedded into optimal cutting temperature compound (Sakura Finetek), gradually frozen in liquid nitrogen via isopentane to avoid tissue damage. Serial sections with a thickness of 6 microns were sliced with a cryostat (Leica). The slides were then stained with a Masson's trichrome staining kit (Sigma-Aldrich) following the manufacturer's instructions. Images of the LV were obtained with 4–6 fields per section (*McMullen et al., 2004*) using a brightfield microscope (Leica). Blue-stained fibrosis areas within sections were determined using colour-based thresholding (*Abràmoff et al., 2004*) and measured with ImageJ software (NIH; http://rsbweb.nih.gov/ij/). The percentage of total fibrosis area was calculated by taking the sum of the blue-stained areas divided by the total LV area.

## Statistics

All experiments and analyses were blinded. Averaged data are presented as means ± standard error of the mean (SEM). The statistical analysis was performed using GraphPad Prism software, version 7.04 (GraphPad). For comparisons between two sets of data, unpaired t-test was used to determine the statistical significance. For comparisons among multiple sets of data with one factor or two factors, one-way or two-way ANOVA was used accordingly, followed by Tukey's post-hoc test. $p < 0.05$ was considered statistically significant.

## Acknowledgements

The authors gratefully acknowledge funding from the National Health and Medical Research Council (NHMRC) of Australia to BM and MF through a project grant (APP1108013), as well as the NHMRC of Australia for a Principal Research Fellowship (APP1135974) and NSW Cardiovascular Disease Senior Scientist Grant to BM. RV, SP, and AP are supported by the BOF Katholieke Universiteit Leuven (TRPLe) and the FWO-Vlaanderen (G0E0317N). CDC is supported by an NSW Health EMCR Fellowship. YG is supported by the Australian Government Research Training Program (RTP). This work is part of a PhD thesis of YG.

## Additional information

### Funding

| Funder | Grant reference number | Author |
|---|---|---|
| National Health and Medical Research Council | APP1108013 | Yang Guo<br>Ze-Yan Yu<br>Charles D Cox<br>Michael P Feneley<br>Boris Martinac |
| National Health and Medical Research Council | APP1135974 | Boris Martinac |
| Fonds Wetenschappelijk Onderzoek | G0E0317N | Silvia Pinto<br>Andy Pironet<br>Rudi Vennekens |
| KU Leuven | TRPLe | Silvia Pinto<br>Andy Pironet<br>Rudi Vennekens |
| NSW Health | EMCRFellowship | Charles D Cox |

The funders had no role in study design, data collection and interpretation, or the decision to submit the work for publication.

## Author contributions
Yang Guo, Data curation, Formal analysis, Validation, Investigation, Methodology, Writing - original draft; Ze-Yan Yu, Conceptualization, Data curation, Formal analysis, Supervision, Investigation, Visualization, Writing - original draft; Jianxin Wu, Data curation, Supervision, Validation, Investigation, Methodology; Hutao Gong, Data curation, Formal analysis, Validation, Investigation, Visualization, Methodology; Scott Kesteven, Formal analysis, Validation, Investigation, Methodology; Siiri E Iismaa, Data curation, Supervision, Methodology; Andrea Y Chan, Silvia Pinto, Data curation, Validation, Investigation, Methodology; Sara Holman, Data curation, Validation, Methodology; Andy Pironet, Data curation, Investigation, Methodology; Charles D Cox, Conceptualization, Supervision, Investigation, Writing - original draft; Robert M Graham, Resources, Funding acquisition, Writing - original draft; Rudi Vennekens, Resources, Funding acquisition, Methodology, Writing - original draft, Project administration; Michael P Feneley, Boris Martinac, Conceptualization, Resources, Supervision, Funding acquisition, Validation, Methodology, Writing - original draft, Project administration

## Author ORCIDs
Boris Martinac (iD) https://orcid.org/0000-0001-8422-7082

## Ethics
Animal experimentation: This study was performed in strict accordance with the recommendations in the Guide for the Care and Use of Laboratory Animals of the National Health and Medical Research Council of Australia. All of the animals were handled according to approved institutional animal care and use. The protocol was approved by the Garvan Institute of Medical Research/St Vincent's Hospital Animal Ethics Committee (Project No. 20_06). All surgery was performed under sodium pentobarbital anesthesia, and every effort was made to minimize suffering.

## Decision letter and Author response
Decision letter https://doi.org/10.7554/eLife.66582.sa1
Author response https://doi.org/10.7554/eLife.66582.sa2

## Additional files

### Supplementary files
• Source data 1. Western blots.

• Supplementary file 1. Haemodynamic and anatomical parameters after 2 days and 14 days of sham/TAC in WT and *Trpm4* cKO mice. Haemodynamic measurements include heart rate (HR), aortic systolic and diastolic pressure, LV systolic pressure, $dP/dt_{max}$ and $dP/dt_{min}$; anatomical measurements include body weight (BW), heart weight (HW), LV weight (LVW), lung weight (LW), tibial length (TL), heart weight normalised by body weight (HW/BW), LV weight normalised by body weight and tibial length (LVW/BW; LVW/TL), lung weight normalised by body weight (LW/BW). Data are presented as means ± SEM. Comparison between sham and TAC in WT or *Trpm4* cKO groups: **$p<0.01$, ***$p<0.001$; Comparison between WT and *Trpm4* cKO TAC groups: #$p<0.05$, ###$p<0.001$.

• Transparent reporting form

### Data availability
All data generated or analysed during this study are included in the manuscript and supporting files. Source data files have been provided for all the figures and tables.

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
