## [Decision Letter]

**Acceptance summary:**

In this work, the authors subjected mice with cardiomyocyte-specific deletion of ion channel TRPM4 to transverse aortic constriction-induced pressure overload. The study demonstrates that cell-specific loss of TRPM4 in cardiomyocytes protects against pathological left ventricular hypertrophy which is associated with an attenuation of pathological changes, including the expression of genes that become dysregulated during the development of pathological hypertrophy. These findings highlight the importance of tissue-specific changes in the pathogenesis of pressure-induced LVH which will further the understanding of heart disease and pathophysiology in the context of heart failure due to arterial hypertension.

**Decision letter after peer review:**

Thank you for submitting your article "The Ca^2+^-activated cation channel TRPM4 is a positive regulator of pressure overload-induced cardiac hypertrophy" for consideration by *eLife*. Your article has been reviewed by 3 peer reviewers, one of whom is a member of our Board of Reviewing Editors, and the evaluation has been overseen by a Senior Editor. The following individuals involved in review of your submission have agreed to reveal their identity: Ken Takahashi (Reviewer #2); Robert Blanton (Reviewer #3).

The reviewers and Editors have discussed the reviews with one another, and this letter to help you prepare a revised submission.

Essential revisions:

This study demonstrates how the TRPM4 ion channel might act as a contributor to the mechanosensory transduction of pressure overload. Using and experimental model, the authors successfully demonstrated that cardiomyocyte-specific deletion of TRPM4 ion channel protects the left ventricle from pressure overload-induced hypertrophy. The conclusions of this paper are mostly supported by data, but some of the wording and data analysis need to be clarified and extended. Improving the following aspects would strengthen the manuscript as described below.

1. While the TRPM4 channel is well known for its association with cardiomyocyte action potential formation and arrhythmia, this manuscript only describes its role in mechanotransduction that was found in this study. The findings of this study may be more comprehensively accepted if there is a hypothesis that links the finding of this study with previous findings.

2. The authors should highlight the differences between this study and the Angiotensin II-induced cardiac hypertrophy Trpm4cKO mice model that has previously been reported (Kecskes et al., Basic Res Cardiol. 2015; 110(4): 43).

3. Signaling studies. Teasing out the effects of TRPM4 deletion on NFAT vs CaMKII signaling would be strengthened by more detailed measures of CaMKII activation state (such as phospho-specific AB to activated CaMKII); NFAT signaling (such as qPCR for RCAN); calcineurin activity or expression levels.

4. Nuclear localization studies. To improve clarity and understanding, the reviewer would prefer for authors to express all of the findings as cytoplasmic/nuclear. Further, would help to do the GAPDH and H2B blots on both the cytoplasmic and nuclear blots, to demonstrate successful fractionation of compartments.

5. Ideally, in cell culture or in vivo, it would be useful to test whether augmentation of CaMKII would restore the normal LV hypertrophy response, which would provide good evidence that CaMKII truly mediates the downstream effects of TRPM4.

---

## [Author Response]

Essential revisions:This study demonstrates how the TRPM4 ion channel might act as a contributor to the mechanosensory transduction of pressure overload. Using an experimental model, the authors successfully demonstrated that cardiomyocyte-specific deletion of TRPM4 ion channel protects the left ventricle from pressure overload-induced hypertrophy. The conclusions of this paper are mostly supported by data, but some of the wording and data analysis need to be clarified and extended. Improving the following aspects would strengthen the manuscript as described below.1. While the TRPM4 channel is well known for its association with cardiomyocyte action potential formation and arrhythmia, this manuscript only describes its role in mechanotransduction that was found in this study. The findings of this study may be more comprehensively accepted if there is a hypothesis that links the finding of this study with previous findings.

We thank the reviewer for this comment and agree that the significance of TRPM4 for cardiac electrical activity is important to mention. On p. 15, line 327-331 we now write:

“Previous studies reported that the TRPM4 current contributes to the mammalian atrial action potential (Simard, Hof, Keddache, Launay, and Guinamard, 2013) as well as to the notch and early repolarization phases of the action potential in Purkinje cells (Hof et al., 2016), providing a potential link to cardiac arrhythmias (Guinamard et al., 2015; Hedon et al., 2021; Wang et al., 2018).”

Moreover, globally speaking the influence of TRPM4 on cardiomyocyte Na^+^ and Ca^2+^ handling will almost certainly contribute to arrhythmia propensity as well as cardiac hypertrophy. Previous work suggests that increased TRPM4 activation may influence arrhythmias through local Na^+^ loading and reverse mode of the N^+^/Ca^2+^ exchanger (NCX) which in turn causes an increase in intracellular Ca^2+^ (Wang et al., 2018). In fact, this mechanism could be shared in TAC-induced hypertrophy and may provide a plausible explanation for how TRPM4 activation influences Ca^2+^ signalling to activate the CaMKII-HDAC-MEF2 pathway (Figure 6). As a result, we have included this putative link between TRPM4 and NCX activity in the Discussion as part of our hypothesis for the involvement of TRPM4 in TAC-induced LVH pathway. However, we should note that this is very distinct from the effects of TRPM4 in angiotensin II-mediated hypertrophy where channel activation is thought to reduce the driving force for store operated Ca^2+^ entry through TRPC1 and TRPC3, as discussed at length in Major Point 2.

2. The authors should highlight the differences between this study and the Angiotensin II-induced cardiac hypertrophy Trpm4cKO mice model that has previously been reported (Kecskes et al., Basic Res Cardiol. 2015; 110(4): 43).

We must admit that we were slightly surprised by this comment from the reviewer because we specifically cited the Kecskes et al. (2015) *Trpm4* cKO study in the Introduction of our original manuscript, and we then devoted approximately half of the Discussion in our original manuscript to highlighting and explaining the differences between the Kecskes et al. (2015) *Trpm4* cKO study and the results of our own study. For your convenience, we have reproduced the relevant sections of our original manuscript below:

Excerpt from the Introduction: (line 111-116)

“Previous studies using *Trpm4* cardiomyocyte-specific knock-out (*Trpm*4 cKO) mice have shown that TRPM4 is a negative regulator of angiotensin II-induced cardiac hypertrophy in mice, which involves the calcineurin-NFAT pathway (Kecskes et al., 2015), and that TRPM4 is essential for survival after myocardial infarction (Hedon et al., 2021; Jacobs et al., 2015). However, whether TRPM4 plays a role in mechanical pressure overload-induced LVH has yet to be determined.”

Excerpt from the Discussion: (line 331-387)

“Importantly, there is evidence that the TRPM4 channel is a critical modulator of ventricular remodelling in cardiac hypertrophy and heart failure (Frede et al., 2020; Jacobs et al., 2015; Kecskes et al., 2015; Mathar et al., 2014). […] Moreover, once activated, CaMKII has been shown to inhibit calcineurin activity (Kreusser et al., 2014; MacDonnell et al., 2009) (Figure 6).”

Clearly, we are arguing that the role of TRPM4 in angiotensin II-induced cardiac hypertrophy is mechanistically distinct from its role in TAC-induced cardiac hypertrophy, and that this mechanistic distinction explains the inhibitory effect of TRPM4 activation on hypertrophy in the former case and the pro-hypertrophic effect of TRPM4 activation in the latter case. In the revised manuscript, we have added the following paragraph to the discussion, which provides a possible mechanism by which TRPM4 is pro-hypertrophic in response to TAC but anti-hypertrophic in response to angiotensin II: (line 388-396)

“The high-frequency/high amplitude Ca^2+^oscillations characteristic of TAC and the consequent intermittent nature of the resulting TRPM4 activation, as distinct from the persistent activation associated with angiotensin II-induced hypertrophy, may mean that TAC-induced TRPM4 activation does not reduce the driving force for Ca^2+^ influx via SOCE that characterises the anti-hypertrophic action of TRPM4 in angiotensin II-induced hypertrophy (see above). Ca^2+^Ca^2+^[…] This would account for the pro-hypertrophic effect of TRPM4 in TAC-induced hypertrophy as distinct from its anti-hypertrophic effect in angiotensin II-induced hypertrophy.”

3. Signaling studies. Teasing out the effects of TRPM4 deletion on NFAT vs CaMKII signaling would be strengthened by more detailed measures of CaMKII activation state (such as phospho-specific AB to activated CaMKII); NFAT signaling (such as qPCR for RCAN); calcineurin activity or expression levels.

CaMKII activation status

Measuring CaMKII activation using a phospho-specific antibody was a great suggestion. We thank the reviewer for this recommendation. We have performed the new western blot experiments using a p-CaMKII antibody, specifically a marker of phosphorylation at threonine 287 which is auto-phosphorylated by CaMKII after activation and thus acts as a marker of CaMKII activation (Lai et al., 1987; Luczak et al., 2020). Briefly, we found that p-CaMKII increased in response to TAC in WT mice and this increased p-CaMKII was abolished in TAC-operated hearts from *Trpm4* cKO mice. Our new results are consistent with our conclusion that the CaMKII-HDAC-MEF2 pathway was activated in response to TAC-induced LV pressure overload. We have provided this data in the new Figure 4. Also, we have added these contents in both the Methods and Results sections.

Calcineurin activity

With due respect to the reviewer’s comments re calcineurin activity and NFAT signalling, the only incontrovertible evidence of calcineurin-NFAT pathway activation is direct measurement of nuclear translocation of NFAT based on measurements of the nuclear and cytoplasmic NFAT fractions, as was done in our current study – please see the review paper by Molkentin, in which he addresses this specific issue, which was cited in our original manuscript [Molkentin, J.D. (2013). Parsing good versus bad signalling pathways in the heart: role of calcineurin-nuclear factor of activated T-cells. Circ Res 113, 16-19]. On the issue of RCAN measurements, in particular, we have recently published our important finding that RCAN (MCIP1) is a particularly poor method of discriminating between calcineurin-NFAT4 pathway activation and CaMKII pathway activation: we observed similar elevations of RCAN (MCIP1) with calcineurin activation alone and with CaMKII activation in the absence of calcineurin activation (see our updated reference to Yu, ZY et al. Cardiac Gq receptors and calcineurin activation are not required for the hypertrophic response to mechanical left ventricular pressure overload. Front Cell Dev Biol. 2021 Feb 15;9:639509, originally cited as appearing in bioRxiv). As explained in our recent paper, the poor specificity of RCAN (MCIP1) is a consequence of the convergence of these two distinct signalling pathways onto a common intra-nuclear complex to initiate hypertrophic transcription and the fact that nuclear export of HDAC4 relieves the repression not only of MEF2 but also of NFAT. Derepression of both MEF2 and NFAT by CaMKII activation results in gene transcription downstream of intra-nuclear NFAT, including expression of RCAN (MCIP1). It follows that increased RCAN gene expression cannot be regarded as a specific marker of calcineurin activation. For the same reason, a similar caveat should apply to conclusions regarding calcineurin activation drawn from experiments based on the NFAT luciferase reporter. In short, we encourage the reviewer to examine our recent publication for further detail, but we are confident that our method of determining calcineurin-NFAT pathway activation and discriminating it from CaMKII pathway activation is superior to the methods suggested by the reviewer. Because we have already published these findings in detail elsewhere, we think it is sufficient to direct the reader to our previous publication (Yu et al., 2021) and the Molkentin (2013) review paper. In the Results section of the revised manuscript, therefore, we have slightly modified the text (yellow highlight) describing the absence of activation of the calcineurin-NFAT pathway after TAC, as follows: (line 286-289)

“These findings indicate that TAC did not result in increased nuclear translocation of NFATc4, the most reliable indicator of calcineurin activation (Molkentin, 2013), and confirm our previous finding (Yu et al., 2021) that the calcineurin-NFAT pathway is not activated by TAC.”

4. Nuclear localization studies. To improve clarity and understanding, the reviewer would prefer for authors to express all of the findings as cytoplasmic/nuclear. Further, would help to do the GAPDH and H2B blots on both the cytoplasmic and nuclear blots, to demonstrate successful fractionation of compartments.

We have seriously considered the reviewer’s suggestion to express all of our findings as cytoplasmic/nuclear ratios. We have now re-analysed our results and provided new data for cytoplasmic/nuclear ratios in Figure 4—figure supplement 2 and Figure 5—figure supplement 1, although we think that these data are not essential to support the main conclusions of our study. The cytoplasmic/nuclear ratio is of mechanistic significance only when there is a *shuttle* of the protein of interest between the nucleus and the cytoplasm, as is the case for HDAC4, but this is not the case for CaMKII, for example.

As requested, to confirm the purity of the fractions extracted from the LV tissue, we have performed the new estern blot experiments using the specific marker proteins: GAPDH for the cytoplasmic fraction and Histone H2B for the nuclear fraction. Each fraction was run side-by-side on the same blot and then probed separately against each of two primary antibodies: GAPDH and Histone H2B to validate the purity of each fraction. Our results demonstrated that we have successfully separated cytoplasmic and nuclear extracts with exceptionally high purity. This result has now been included in the Results section, and we have also provided the blot images in Figure 4—figure supplement 1.

5. Ideally, in cell culture or in vivo, it would be useful to test whether augmentation of CaMKII would restore the normal LV hypertrophy response, which would provide good evidence that CaMKII truly mediates the downstream effects of TRPM4.

We thank the reviewer for this suggestion, but we believe such an undertaking would require an extensive amount of work that would warrant a separate study. This is because in order to ‘augment’ CaMKII, we envisage two options: (1) a genetic approach (cardiomyocyte over- expression) or (2) a pharmacological approach (selective CaMKII activator).

Regarding the genetic approach, we would require a new inducible transgenic mouse model. Even if we were to attempt the same type of experiment in vitro and develop a plasmid-based approach to over-express CaMKII, in combination with electroporation or viral transduction, we have no method in vitro to mechanically stimulate adult cardiomyocytes or cardiomyocyte-like cell lines to induce hypertrophy to mimic our surgical mouse model.

Regarding the pharmacological approach, no selective CaMKII activators exist. There are CaMKII selective inhibitors such as KN-93 and this molecule even has an inactive derivate, KN-92, that can be used as a control but here we must point out that extensive studies have already documented a key role for CaMKIIδ in pressure overload induced hypertrophy using knockout mouse models. Particularly Kreusser et al., 2014 Circulation provide compelling evidence that CaMKIIδ is essential for pressure overload induced hypertrophy using a similar TAC model (see Kreusser et al., 2014 Figure 3c). This publication and others that support the role of CAMKII in TAC-induced hypertrophy are discussed at length in our previous publication: Yu et al. (2021). Frontiers in Cell and Developmental Biology.

In summary, because CaMKII is indispensable for the hypertrophic response to TAC and that nuclear export of HDAC4 is reduced in *Trpm4* cKO mice, we believe our data strongly supports the presented model that TRPM4 is part of the mechanosensory machinery that drives the CaMKII-HDAC4-MEF2 pathway that is selectively activated under pressure overload conditions to induce cardiac hypertrophy (Yu et al., 2021, Frontiers in Cell and Developmental Biology).